# Docosahexaenoic Acid as the Bidirectional Biomarker of Dietary and Metabolic Risk Patterns in Chinese Children: A Comparison with Plasma and Erythrocyte

**DOI:** 10.3390/nu14153095

**Published:** 2022-07-28

**Authors:** Zhi Huang, Ping Guo, Ying Wang, Ziming Li, Xiaochen Yin, Ming Chen, Yong Liu, Yuming Hu, Bo Chen

**Affiliations:** 1Key Laboratory of Phytochemical R&D of Hunan Province, School of Chemistry & Chemical Engineering, Hunan Normal University, Lu Mountain Road No. 286, Changsha 410081, China; hhsfbyzhige@hotmail.com (Z.H.); gpguoping@126.com (P.G.); wy792722290@163.com (Y.W.); liuyong@hunnu.edu.cn (Y.L.); 2School of Public Health and Laboratory, Hunan University of Medicine, Jinxi Road No. 492, Huaihua 418000, China; 3The Department of Toxicology, Hunan Provincial Center for Disease Control and Prevention, Furong Road No. 450, Changsha 410005, China; hncdclzm@163.com (Z.L.); yinxiaochen83@sina.com (X.Y.); xiaomingzichen1990@163.com (M.C.)

**Keywords:** docosahexaenoic acid, plasma, erythrocyte, dietary patterns, metabolic risk patterns

## Abstract

Objective: The present study aims to measure docosahexaenoic acid (DHA) in both the plasma and erythrocyte of a child population and compares them with respect to their associations with dietary and metabolic risk patterns. Methods: A cross-sectional study was conducted, and a total of 435 children ages 5–7 years old were recruited. Diet information was collected using a food frequency questionnaire (FFQ). The physical indicators, blood pressure, and glycolipid metabolic indicators were determined. The plasma and erythrocyte DHA were analyzed using a gas chromatography mass spectrometer. Principal component analysis was used to identify dietary and metabolic risk patterns. Multivariate regression analyses were used to investigate the associations of DHA status with dietary and metabolic risk patterns. Results: A significant correlation between plasma and the erythrocyte DHA concentration was found (r = 0.232, *p* < 0.001). A diversified dietary pattern characterized that a high intake of diversified foods had a positive association with the plasma DHA level (β = 0.145, 95% CI: 0.045~0.244, *p* = 0.004). Children of obesity risk patterns with a high weight, pelvis breadth, BMI, upper arm circumference, and chest circumference had lower plasma DHA concentrations (OR = 0.873, 95% CI: 0.786~0.969, *p* = 0.011). Children with higher plasma and erythrocyte DHA concentrations were adhered to blood lipid risk patterns with high CHOL and LDL-C levels. The plasma DHA (OR = 1.271, 95% CI: 1.142~1.415, *p* < 0.001) had a stronger association with a blood lipid risk pattern than erythrocyte (OR = 1.043, 95% CI: 1.002~1.086, *p* = 0.040). Conclusions: The diversified dietary pattern had a higher plasma DHA concentration. Lower levels of plasma DHA were positively associated with obesity in children. DHA in plasma appears to be more strongly associated with blood lipid metabolism than erythrocyte. Plasma DHA may be a more sensitive bidirectional biomarker to evaluate the recently comprehensive diet intake and metabolic risk of children.

## 1. Introduction

Childhood is an important period for rapid growth and development. Fatty acids (FAs) play an important role in energy metabolism and compose fundamental structures through various types of lipids [1]. Important among these structures are *n*-3 (omega-3) polyunsaturated fatty acids (PUFAs), which are involved neurodevelopment, immune function, cellular signaling processes, angiogenesis, and inflammatory response [2,3,4], playing a crucial role in children’s growth, development, and function. Docosahexaenoic acid (DHA, C22:6*n*-3) is the predominant *n*-3 PUFA, making up over 90% of the *n*-3 PUFAs in the brain and 10–20% of its total lipids [5]. DHA status leads to many positive physiological and behavioral effects in children. Several previous studies confirmed that the intake or circulating levels of DHA are associated with children’s health, such as metabolic [6], immune [7], and central nervous system disorders [8]. In addition, diversified preformed DHA via dietary and supplemental sources was explored to maintain optimal levels of DHA in populations of children [9,10].

The sources of human DHA are dependent on the exogenous diet intake or endogenous synthesis in the organism [11]. However, the capacity to synthesize DHA is limited in humans. An adequate supply of DHA mainly depends on diet intake [12]. Traditional diet assessment methods for DHA intake rely on the accuracy of memories, awareness of fat intake, or willingness to report the details of a diet, for which there is inevitable recall bias [13]. Biomarkers of DHA can adequately reflect temporal changes in diet DHA intake and be measured in various readily available tissues. The DHA in various blood fractions and tissues (e.g., plasma or serum, erythrocytes, and adipose tissue) are widely used as biomarkers to assess dietary intake [14]. Nevertheless, the majority of previous studies only focused on the DHA intake of individual foods. The potential interactions between total dietary intake and foods are neglected because dietary intake is a combination of diverse foods. Dietary pattern analysis investigating a comprehensive diet has been developed by many researchers to assess the association of the whole diet with health [15,16,17,18], whereas few studies reported the associations of dietary patterns with the DHA biomarker [19,20].

Subsequently, the DHA biomarker was widely used in epidemiologic studies to predict the risk of metabolic diseases [21,22]. The Omega-3 Index (EPA + DHA) was proposed to be considered a new risk factor for death from coronary heart disease (CHD) [23]. In addition, the DHA biomarker was also reported to be associated with the metabolic risk of children (e.g., obesity [24] and diabetes [25]) or the risk of developing these diseases in adulthood of children [26]. However, the potential interactions of metabolic risk variables (e.g., blood glucose, blood lipid, blood pressure, or obesity) were also ignored.

Adipose tissue (AT), as the major storage site of lipids, can provide a stable and valid estimate of DHA status [27]. However, this tissue is not suitable for most epidemiologic studies, because the acquisition of tissue has a more invasive risk. Blood offered a less invasive opportunity, and it is more available and widely used. The DHA status of plasma is a reliable indicator to evaluate diet intake [28]. Several studies considered that plasma DHA only reflects the dietary lipid intake in the short term [29,30,31]. In addition, other studies reported that DHA in erythrocyte might be a more excellent candidate biomarker to explore the association with human diseases than other blood fractions, because erythrocyte reflects not only information related to the diet intake of an individual but also information related to FA endogenous metabolism [6]. Several studies reported on the comparison of plasma and erythrocyte DHA status in pregnancy and lactating women and its association with DHA intake, but few studies have been focused on the comparison of plasma and erythrocyte DHA status in children [32,33].

Metabolites to relate diet intakes and health outcomes as bidirectional biomarkers can be used to evaluate how this set of diet-derived biomarkers is associated with disease risk and indicate entry points for precision prevention and intervention strategies [34]. Therefore, a cross-sectional study is conducted here. We aim to measure DHA in both the plasma and erythrocyte of a population of children and compare them with respect to their associations with dietary and metabolic risk patterns. We hope to provide a precisely bidirectional biomarker of blood DHA to relate diet intakes to the health outcomes of children in the future (Figure 1).

## 2. Materials and Methods

### 2.1. Study Design and Study Population

A cross-sectional study was conducted in Xiangtan, a city in Hunan province located in south central China. All children aged 5–7 years old in a primary school in Xiangtan were invited to participate. Total of 435 children were recruited. Of them, 21 were excluded due to incomplete dietary assessment or no collected blood samples.

### 2.2. Dietary Pattern Assessment

Diet information was collected using a validated food frequency questionnaire (FFQ) as previously described in [35]. The estimated portion size expressed in grams or milliliters and the frequency of 55 food items for children over the previous 12 months were asked to be recalled by their caregivers. The mean daily intake of each food item was calculated using the estimated portion size and frequency.

Principal component analysis was used to identify dietary patterns. Nineteen food groups were further concluded from 55 food items and included in the principal component analysis. The number of dietary patterns was identified based on the eigenvalue, scree plots, factor interpretability, and explained variance. Factor loadings represented the correlation between food items and dietary patterns. Factor loadings > |0.3| were considered to contribute significantly to the dietary pattern. Aside from that, factor scores summing the intakes of each food group weighted according to factor loadings were calculated for each pattern and each individual. The factor scores were further categorized into four quartiles (Q1–Q4). Q1 represented a weak association with the dietary pattern, and Q4 represented a strong association with the dietary pattern.

The results of the principal component analysis and factor loadings for dietary patterns were reported in our previous study [35]. The diversified patterns, plant patterns, and beverage and snack patterns were identified in this population. The diversified pattern is characterized by high intakes of diversified foods, such as fruits, nuts, leafless vegetables, poultry, fungi and algae, fresh beans, tubers, fish, meat, soybeans and their products, snacks, rice, shrimp, crab, and shellfish. The plant pattern had high intakes of coarse cereals, soybeans and their products, leafless vegetables, and tubers and low intakes of poultry and meat. The beverage and snack pattern involves consuming a high proportion of beverages, snacks, and milk and its products and low intakes of shrimp, crab, shellfish, and fish (Appendix A).

### 2.3. DHA Analysis

Venous whole-blood samples after overnight fasting were collected from the cubital vein using a metal-free and heparinized vacuum tube. The whole-blood samples were centrifuged (3000 rpm for 10 min) to separate the erythrocytes and plasma. These samples were stored at −80 °C after separation.

For the erythrocyte DHA, the detailed analysis procedure was reported in our previous study [36]. Twenty-seven fatty acids (including DHA) were determined in the erythrocyte in the previous study. Briefly, an internal standard working solution (IS) and 1 mL 3 N methanolic hydrochloric acid were added to 50 µL of erythrocyte and heated at 90 °C for 1.5 h for transmethylation. Two milliliters of hexane was added, vortexed, and separated. The hexane layer was transferred and concentrated by nitrogen blowing. One milliliter of hexane was dissolved. Finally, the samples (1 µL) were analyzed using a gas chromatography mass spectrometer (GCMS-QP2010, Shimadzu Corp., Kyoto, Japan) and separated using an HP-88 column (dimensions: 100 m × 0.25 mm × 0.20 µm; Agilent Technologies, Santa Clara, CA, USA). Quantification was based on calibration with methyl nonadecanoate (Aladdin, Shanghai, China) as the IS. The results were expressed in µg/mL.

For the plasma DHA, 100 µL of plasma was pipetted into a 4-mL propylene pipe. A small amount of anhydrous sodium sulfate and 1 mL IS dissolved in isooctane were added, and then 200 μL of a 0.5-mol/L potassium hydroxide/methanol solution was added. The tubes were capped and heated to 65 °C for 15 min and then cooled and centrifuged. The upper layer (isooctane) was analyzed using a gas chromatography mass spectrometer (GCMS-QP2010) and separated using an SH-Rxi-5Sil MS column (30 m × 0.25 mm × 0.25 µm). The injector was maintained at 250 °C, and the detector was maintained at 230 °C. The temperature program was as follows: an initial oven temperature setting of 200 °C for 2 min, an increase of 10 °C/min to 250 °C, and holding for 3 min, followed by another increase of 30 °C/min to 280 °C and holding for 2 min. Helium was used as the carrier gas. The total analysis time was approximately 13 min. Quantification was based on calibration with methyl docosanoic acid as the IS. To evaluate the recovery, three concentrations of DHA were added to the plasma samples, and the sample recovery rates were 89.75–100.65%. The results were expressed in µg/mL.

### 2.4. Metabolic Risk Variables

Metabolic risk variables including weight, height, sitting height, chest circumference, upper arm circumference, shoulder width, pelvis breadth, blood pressure (BP), blood glucose (GLU), triglyceride (TG), cholesterol (CHOL), high-density lipoprotein cholesterol (HDL-C), and low-density lipoprotein cholesterol (LDL-C) levels were assessed in this study. Weight (kg) and height (cm) were measured using an electronic instrument. The body mass index (BMI) was calculated as the weight (kg) divided by height squared (m^2^). The sitting height, chest circumference, upper arm circumference, shoulder width, and pelvis breadth were measured by a meter ruler and expressed in cm. BP was measured by a mercury sphygmomanometer and expressed in mmHg. The GLU, TG, CHOL, HDL-C, and LDL-C levels were determined from the serum with an AU680 clinical chemistry analyzer (Beckman Coulter, Brea, CA, USA) and diagnostic kits (Fosun, Shanghai, China).

The metabolic risk patterns were identified by principal component analysis. The weight, height, BMI, sitting height, chest circumference, upper arm circumference, shoulder width, pelvis breadth, and BP, GLU, TG, CHOL, HDL-C, and LDL-C levels were included in the principal component analysis. The number of metabolic risk patterns was identified, referring to the dietary pattern analysis. Factor loadings represented the correlation between the metabolic risk variables and metabolic risk patterns. Factor loadings >|0.3| were considered to contribute significantly to the metabolic risk patterns. The factor scores summed each metabolic risk variable weighted according to the factor loadings. They were then calculated and further categorized into four quartiles (Q1–Q4). Q1 represented a weak association with the metabolic risk pattern, and Q4 represented a strong association with the metabolic risk pattern.

### 2.5. Other Related Variables

The demographic variables collected for children including age (4–5 years or 6–7 years), sex (boy or girl), caregiver group (parents or grandparents and others), caregiver occupation (public institution, non-public institution, or unemployed), caregiver education (junior and below, senior, or college and above), and family economic level (family annual income (in CNY) <20,000, 20,000–50,000, or ≥50,000).

### 2.6. Statistical Analysis

The continuous variables with normally distributed data were expressed by the mean (standard deviation (SD)) or median (interquartile range (IQR)) with skewed data. The categorical variables were expressed by numbers and percentages. Correlations of the related variables were conducted by Pearson’s correlation analysis for the normally distributed data or Spearman’s correlation analysis. Comparisons of the DHA statuses in the four quartiles of dietary and metabolic risk patterns were conducted by variance analysis for the normally distributed data or the Kruskal–Wallis test. Multivariate linear regression analyses were used to investigate the associations of the DHA status with dietary patterns. The plasma and erythrocyte DHA were dependent variables. Model 1 was unadjusted. Model 2 was adjusted for the demographic variables (age, sex, caregiver, caregiver’s eduction and occupation, and family economic level). Standardized coefficients (β) with a 95% confidence interval (CI) were calculated to determine the strength of the associations. Multivariate logistic regression analyses were used to investigate the associations of the DHA status with metabolic risk patterns. Metabolic risk patterns were the dependent variables. Model 1 was unadjusted. Model 2 was adjusted for the demographic variables. Model 3 was adjusted for demographic variables and diet intake. An odds ratio (OR) with a 95% CI was calculated to determine the strength of the associations, where *p* < 0.05 was considered indicative of statistical significance. The statistical analysis was performed using SPSS software (version 13.0, Chicago, IL, USA).

### 2.7. Ethics Approval

All subjects provided informed consent for inclusion before participating in the study. The study was approved by the Ethics Committee of the Hunan Provincial Center for Disease Control and Prevention (HNCDC-BJ20190003).

## 3. Results

### 3.1. Participants’ Characteristics

The demographic characteristics and metabolic risk variables of children are shown in Table 1. The medians (IQR) of the plasma and erythrocyte DHA concentrations were 7.91 (6.22, 10.45) and 13.89 (7.49, 18.99) μg/mL, respectively. There was a significant correlation between the plasma and erythrocyte DHA concentrations among children (r = 0.232, *p* < 0.001) (Figure 2).

### 3.2. Correlation between DHA and Dietary Patterns

The correlations of food items with the plasma and erythrocyte DHA levels among children are indicated in Table 2. The intake of animal food products, including meat, poultry, eggs, and fish had significant positive correlations with the plasma DHA, especially for eggs (r = 0.225, *p* < 0.001). Aside from that, coarse cereals, fungi and algae, and fruits were also positively associated with the plasma DHA. However, there were no significant correlations between the erythrocyte DHA and intake of food items except beverages (r = − 0.138, *p* = 0.005).

Figure 3 presents the plasma and erythrocyte DHA status in four quartiles of factor scores among the dietary patterns. The higher factor scores had higher plasma DHA concentrations for the diversified pattern (χ^2^ = 9.845, *p* = 0.020). There was a significant difference in plasma DHA concentration in the four quartiles of the plant pattern (χ^2^ = 11.627, *p* = 0.009) (Figure 3a). No significant differences for the erythrocyte DHA concentration were found in the dietary patterns (Figure 3b).

The association of dietary patterns with the plasma and erythtocyte DHA levels by multivariate linear regression analysis are displayed in Table 3. The diversified pattern had a positive association with the plasma DHA level (β = 0.145, 95% CI: 0.045~0.244, *p* = 0.004) after adjusting for age, sex, caregiver, caregiver’s education and occupation, and family economic level. The beverage and snack pattern was weakly negatively related to the plasma DHA level (β = −0.092, 95% CI: −0.187~0.003, *p* ≈ 0.05). Moreover, no significant associations of the erythrocyte DHA level with dietary patterns were found

### 3.3. Correlation between DHA and Metabolic Risk Patterns

Table 4 showed the coefficients of correlation between plasma and erythrocyte DHA and the metabolic risk indicators. The physical indicators including weight (r = −0.163, *p* = 0.001), height (r= −0.153, *p* = 0.002), BMI (r = −0.097, *p* = 0.049), sitting height (r = −0.146, *p* = 0.003), chest circumference (r = −0.118, *p* = 0.017), shoulder width (r = −0.170, *p* = 0.001), and pelvis breadth (r= −0.142, *p* = 0.004) were significantly negatively correlated with the plasma DHA. CHOL (r = 0.269, *p* < 0.001) and LDL-C (r = 0.269, *p* < 0.001) were positively correlated with the plasma DHA. No significant correlations were found between the metabolic risk indicators and erythrocyte DHA except the upper arm circumference (r = −0.139, *p* = 0.005) and CHOL (r = 0.120, *p* = 0.014).

Three metabolic risk patterns were identified in this study. The factor loadings of each pattern are shown in Table 5. Pattern 1 had high weight, pelvis breadth, BMI, upper arm circumference, and chest circumference associated with a high risk of obesity, named the obesity risk pattern. Pattern 2 was regarded as the blood lipid risk pattern, characterized by high CHOL and LDL-C levels. Pattern 3 was labeled the blood pressure risk pattern, characterized by high SBP and DBP. The higher factor scores had lower plasma DHA concentrations for the obesity risk pattern (χ^2^ = 9.764, *p* = 0.021) but higher concentrations for the blood lipid risk pattern (χ^2^ = 27.848, *p* < 0.001) (Figure 4a). That aside, there was a significant difference in the erythrocyte DHA concentrations in the four quartiles of the obesity risk pattern (χ^2^ = 13.667, *p* = 0.003) (Figure 4b).

Children with higher plasma and erythrocyte DHA concentrations were adhered to the blood lipid risk pattern by logistic regression analysis. Plasma DHA (OR = 1.271, 95% CI: 1.142~1.415, *p* < 0.001) had a stronger association with the blood lipid risk pattern than erythrocyte (OR = 1.043, 95% CI: 1.002~1.086, *p* = 0.040). No significant associations were found between the obesity risk pattern and plasma and erythrocyte DHA after adjusting for demographic variables. However, the children in the obesity risk pattern had lower plasma DHA concentrations (OR = 0.873, 95% CI: 0.786~0.969, *p* = 0.011) after adjusting for demographic variables and the intake of meat, poultry, eggs, and fish. Aside from that, there were no significant associations between the blood pressure risk pattern and DHA concentrations of the plasma and erythrocyte (Table 6).

## 4. Discussion

The evaluation of DHA status has been widely focused on because of application of *n*-3 FA supplementation in functional food [37] and the management and treatment of diseases [38,39]. In this study, we observed a significant positive correlation between plasma and erythrocyte DHA. Moreover, animal food products and the diversified pattern had associations with the plasma DHA level. In addition, children in the obesity risk pattern had lower plasma DHA concentrations. The plasma and erythrocyte DHA concentrations had associations with the blood lipid risk pattern of children, and there was a stronger association in plasma than in erythrocyte.

In this study, only the plasma DHA had associations with diet intake, and no significant correlations of erythrocyte DHA with diet intake (except for beverages) were found. This result is consistent with previous studies indicating that plasma DHA is associated with diet intake [29,30,31]. For another matter, Sun et al.’s study considered erythrocyte DHA as more strongly correlated with diet intake than plasma [14], but their study focused on foods of marine origin enriched DHA. Meyer et al. confirmed that the dose was a strong predictor for circulating levels of DHA [40]. However, the intake of marine food for our population was low, and the plasma DHA had a strong correlation with animal food products in this study, especially for eggs. Eggs with a higher content of phospholipids offered enriched DHA, and its DHA was easily absorbed due to the high DHA content of sn-2 in phospholipids [41,42]. Furthermore, there were different dietary assessment methods, population characteristics, and variations in intake. These reasons may disturb the correlation of erythrocyte DHA, reflecting long-term dietary intake with the diet in this study.

Moreover, the diversified pattern had a positive association with the plasma DHA level, but the beverage and snack pattern was weakly negative in relation to the plasma DHA level. The diversified pattern, with high intakes of fruits, nuts, leafless vegetables, poultry, fungi and algae, fresh beans, tubers, fish, meat, soybeans and their products, snacks, rice, shrimp, crab, and shell fish, included diversified foods, an approach to the oriental healthy pattern which is recommended by Chinese dietary guidelines [43]. The beverage and snack pattern is characterized by high intakes of beverages and snacks, which may increase metabolic risk (e.g., obesity [44]). Our previous studies reported that the dietary pattern represents the overall diet intake [45,46]. These results implied that plasma DHA may become a biomarker to evaluate the recent quality of comprehensive diet intake.

Few studies focused on the DHA status’s effect on the metabolic risk patterns. In this study, we found three metabolic risk patterns in children. No significant associations were found between the obesity risk pattern and plasma and erythrocyte DHA, adjusting for the demographic characteristics. The previous studies demonstrated that children with obesity presented lower DHA statuses for erythrocyte [6,47]. However, children in the obesity risk pattern had a significant association with the plasma DHA concentration after adjusting for demographic variables and the intake of animal food products in this study. There was an interaction between diet intake and DHA status and obesity in children. Furthermore, many epidemiology studies have shown that DHA status is inversely associated with blood pressure [48,49]. However, no significant association was found between DHA status and the blood pressure risk pattern, represented by high SBP and DBP values.

Aside from that, we found that the blood lipid risk pattern with high CHOL and LDL-C levels was positively associated with the concentrations of plasma and erythrocyte DHA. Previous studies showed that a high DHA supplement intake increased the plasma concentrations of the total and LDL cholesterol [50,51]. These studies considered that the reason for this may be an increased intake of fat and cholesterol contained in DHA supplements. In this study, the intake of animal food products had a significant positive correlation with plasma DHA. These results hint that the intake of DHA mainly originated from animal food products with high cholesterol levels in this population, which may increase the risk of lipid metabolic disorders.

On the other hand, many studies held that erythrocyte maintained a more stable FA composition compared with the plasma FA levels, which may be a more excellent candidate biomarker to reflect individual dietary intake and endogenous metabolism [52]. However, there was a stronger association by the lipid risk pattern with the DHA in plasma than in erythrocyte in our study. That aside, Patel et al. also found that the associations of diabetes were greater in magnitude with plasma FAs than with erythrocyte FAs [53]. Their study considered that the stronger associations may be explained by the difference in fatty acid composition in the blood fraction. In this study, a lower level of DHA in plasma (7.91 μg/mL) was found than in erythrocyte (13.89 μg/mL), which may be more easily impacted by other factors. These results hint that plasma may be a more sensitive blood fraction for epidemiologic studies of association with metabolic risk in children. However, the stronger findings for the plasma DHA should be confirmed in larger studies and in different populations.

This is the first study to report the comparative assessment of plasma and erythrocyte DHA as bidirectional biomarkers of dietary and metabolic risk patterns in Chinese children. The results of the present study provide strong evidence for the precise selection of DHA biomarkers in epidemiological studies. However, the limitation of this study is the cross-sectional design, and the causality of these associations needs to be explored in further studies. Second, different methods of transmethylation in DHA analysis may have affected the comparison of the plasma and erythrocyte DHA statuses. Lastly, a majority of the studies reported that circulating levels of DHA were good biomarkers of marine food intake [54,55]. In this case, the subjects were children from areas with low marine food intake, and the association of plasma DHA with dietary patterns might be difficult to use with children from other areas.

## 5. Conclusions

In conclusion, the diversified dietary pattern had a higher plasma DHA concentration. Lower levels of plasma DHA were positively associated with obesity in children. DHA in plasma appears to be more strongly associated with blood lipid metabolism than DHA in erythrocyte. Plasma DHA may be a more sensitive bidirectional biomarker to evaluate recently comprehensive diet intake and metabolic risk in children.

## Figures and Tables

**Figure 1 nutrients-14-03095-f001:**
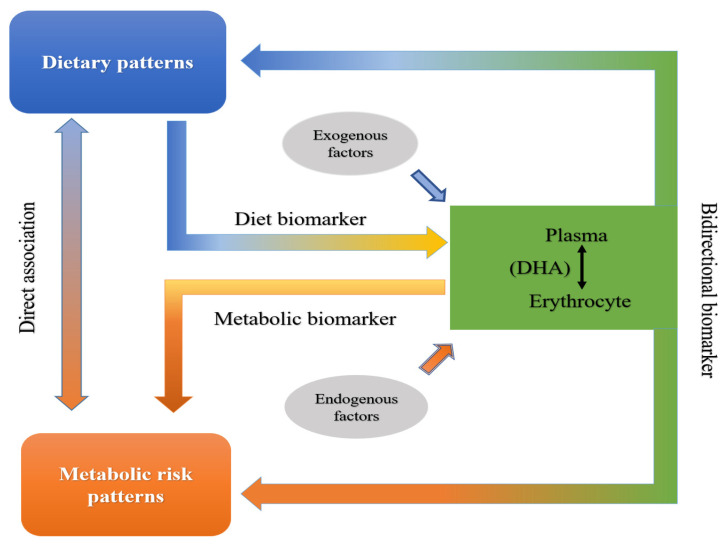
Schematic diagram to precisely identify bidirectional biomarker of blood DHA.

**Figure 2 nutrients-14-03095-f002:**
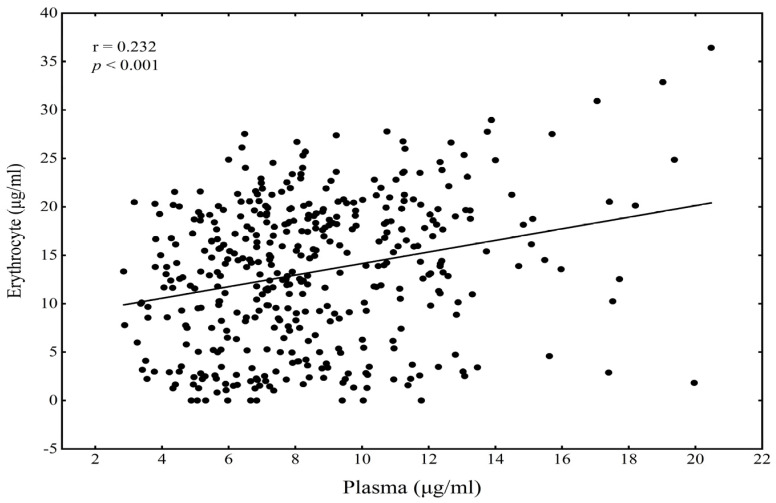
The correlation of plasma DHA with erythrocyte DHA among children.

**Figure 3 nutrients-14-03095-f003:**
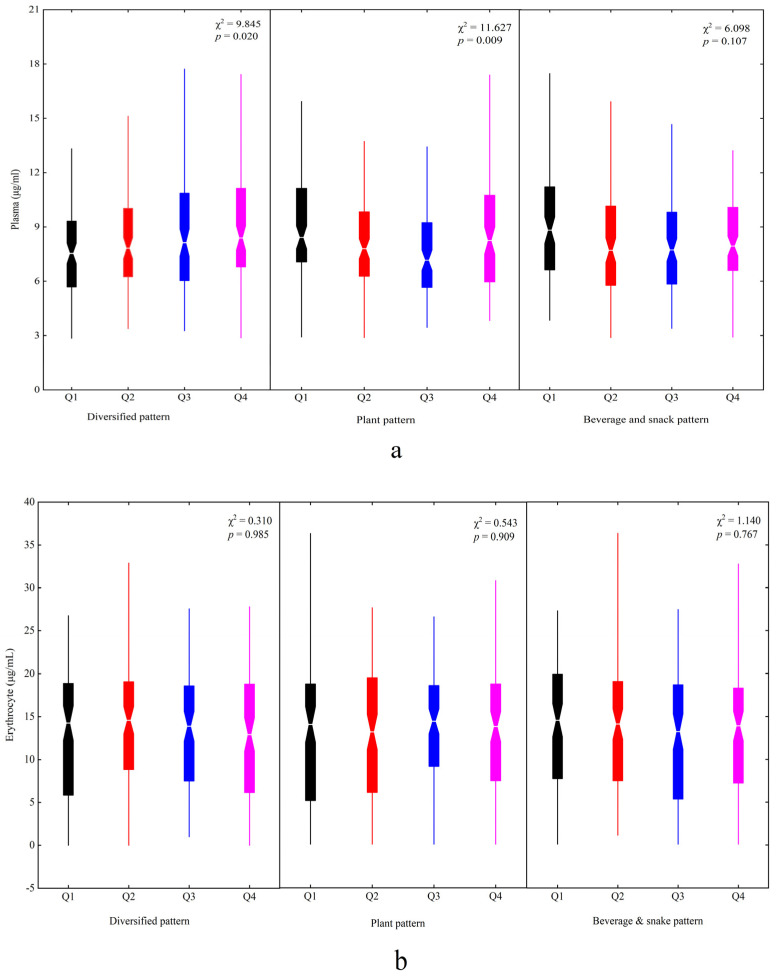
The plasma and erythrocyte DHA status in different dietary patterns among children: (**a**) plasma and (**b**) erythrocyte. Q1~Q4 represented four quartiles of factor scores for dietary patterns: Q1, black; Q2, red; Q3, blue; Q4, magenta.

**Figure 4 nutrients-14-03095-f004:**
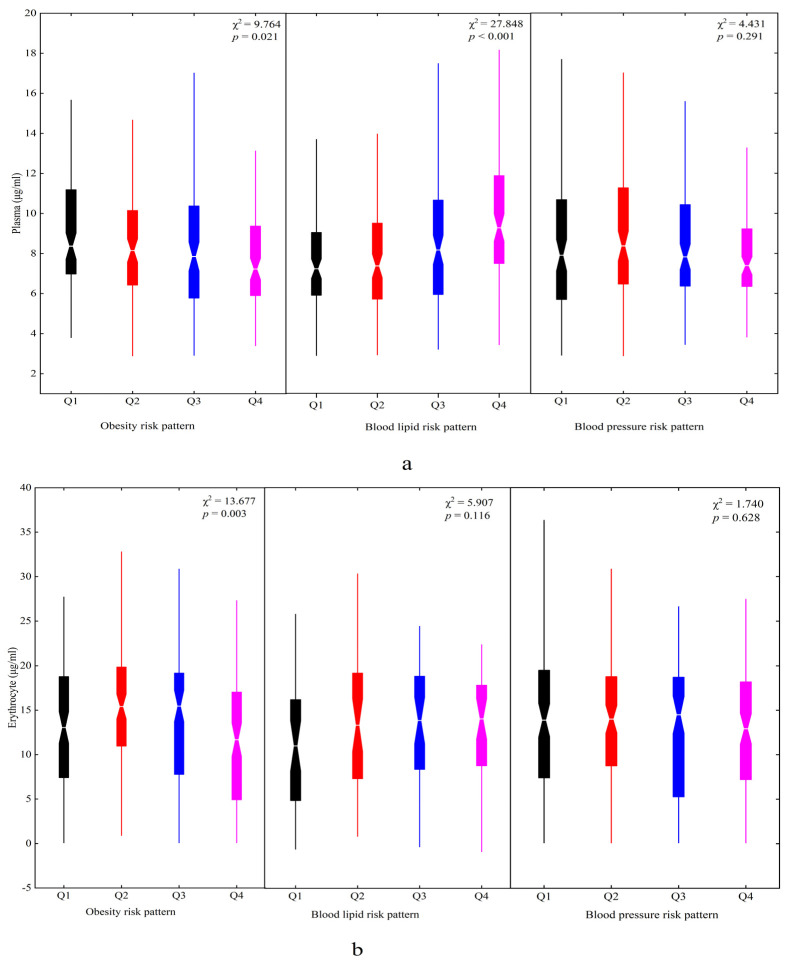
The plasma and erythrocyte DHA status in different metabolic risk patterns among children: (**a**) plasma and (**b**) erythrocyte. Q1~Q4 represented four quartiles of factor scores for dietary patterns: Q1, black; Q2, red; Q3, blue; Q4, magenta.

**Table 1 nutrients-14-03095-t001:** The demographic characteristics and metabolic risk variables of children (*N* = 414).

Indices	*N* (%) or Median (IQR)
Demographic Variables	
Age	
4–5 years	202 (48.79)
6–7 years	212 (51.21)
Gender	
Boy	217 (52.42)
Girl	197 (47.58)
Caregiver	
Parents	238 (57.49)
Grandparents and others	176 (42.51)
Caregiver’s occupation	
Public institution staff	35 (8.45)
Non-public institution staff	150 (36.23)
Unemployment	229 (55.31)
Caregiver’s education	
College and above	36 (8.70)
Senior	115 (27.78)
Junior and below	263 (63.53)
Family economic level	
CNY 50,000 and above	128 (30.92)
CNY 20,000–50,000	183 (44.20)
Below CNY 20,000	103 (24.88)
Metabolic risk variables	
Physical indicators	
Weight (kg)	19 (17, 21)
Height (cm) *	114 (6)
BMI	14.75 (13.96, 15.69)
Sitting height (cm) *	63 (3)
Chest circumference (cm)	54 (52, 56)
Upper arm circumference (cm)	16 (16, 17)
Shoulder width (cm)	31 (29, 32)
Pelvis breadth (cm)	57 (54, 59)
Blood pressure	
SBP (mm/Hg)	91 (85, 97)
DBP (mm/Hg)	58 (54, 62)
Glycolipid metabolic indicators	
GLU (mmol/mL)	4.85 (4.53, 5.30)
TG (mmol/mL)	0.94 (0.67, 1.39)
CHOL (mmol/mL)	4.13 (3.71, 4.53)
HDL-C (mmol/mL)	1.54 (1.35, 1.73)
LDL-C (mmol/mL)	2.14 (1.81, 2.49)
DHA (μg/mL)	
Plasma DHA	7.91 (6.22, 10.45)
Erythrocyte DHA	13.89 (7.49, 18.99)

* Mean (SD).

**Table 2 nutrients-14-03095-t002:** The correlation of food items with plasma and erythrocyte DHA status among children.

Food Items	Plasma DHA		Erythrocyte DHA	
	r	*p*	r	*p*
Rice	−0.078	0.114	−0.063	0.201
Wheat flour	0.052	0.293	0.033	0.497
Coarse cereals	0.100	0.042	0.072	0.144
Tubers	0.050	0.308	−0.001	0.977
Soybean and its products	0.056	0.260	−0.004	0.929
Meat	0.166	0.001	−0.072	0.143
Poultry	0.152	0.002	−0.034	0.493
Eggs	0.225	<0.001	0.091	0.064
Fish	0.141	0.004	0.066	0.181
Shrimp, crab, and shellfish	0.074	0.134	0.029	0.555
Milk and its products	−0.005	0.927	0.084	0.088
Leafy vegetable	0.074	0.134	0.039	0.429
Leafless vegetable	0.022	0.651	0.040	0.417
Fresh beans	0.048	0.334	−0.020	0.683
Fungi and algae	0.109	0.027	0.030	0.545
Fruits	0.101	0.040	0.013	0.796
Beverage	−0.062	0.211	−0.138	0.005
Nuts	0.088	0.075	0.023	0.635
Snacks	−0.038	0.438	−0.035	0.472

**Table 3 nutrients-14-03095-t003:** The association of dietary patterns with plasma and erythtocyte DHA status by multivariate linear regression analysis.

Dietary Patterns	Plasma		Erythrocyte	
	β (95% CI)	*p*	β (95% CI)	*p*
Diversified pattern				
Model 1	0.165 (0.070, 0.261)	0.001	−0.002 (−0.099, 0.095)	0.967
Model 2	0.145 (0.045, 0.244)	0.004	−0.008 (−0.110, 0.094)	0.875
Plant pattern				
Model 1	−0.068 (−0.165, 0.029)	0.167	0.024 (−0.073, 0.121)	0.622
Model 2	−0.075 (−0.171, 0.021)	0.125	0.018 (−0.080, 0.116)	0.725
Beverage and snack pattern				
Model 1	−0.110 (−0.207, −0.014)	0.025	−0.032 (−0.128, 0.065)	0.520
Model 2	−0.092 (−0.187, 0.003)	0.057	−0.031 (−0.128, 0.066)	0.531

Model 1: unadjusted; model 2: adjusted for age, sex, caregiver, caregiver’s eduction and occupation, and family economic level.

**Table 4 nutrients-14-03095-t004:** The correlation of metabolic risk variables with plasma and erythtocyte DHA status among children.

Metabolic Risk Variables	Plasma DHA		Erythrocyte DHA	
	r	*p*	r	*p*
Weight	−0.163	0.001	−0.076	0.122
Height	−0.153	0.002	−0.046	0.355
BMI	−0.097	0.049	−0.071	0.147
Sitting height	−0.146	0.003	−0.004	0.939
Chest circumference	−0.118	0.017	−0.093	0.057
Upper arm circumference	−0.078	0.112	−0.139	0.005
Shoulder width	−0.170	0.001	−0.017	0.736
Pelvis breadth	−0.142	0.004	−0.066	0.182
SBP	0.011	0.817	−0.061	0.217
DBP	−0.042	0.392	−0.049	0.321
GLU	0.003	0.958	−0.094	0.056
TG	0.057	0.251	−0.001	0.990
CHOL	0.269	<0.001	0.120	0.014
HDL-C	0.011	0.822	0.075	0.129
LDL-C	0.269	<0.001	0.069	0.162

**Table 5 nutrients-14-03095-t005:** Factor loadings for metabolic risk patterns.

Metabolic Risk Variables	Obesity Risk Pattern	Blood Lipid Risk Pattern	Blood Pressure Risk Pattern
Weight	0.980	−0.029	−0.073
Height	0.740	−0.083	−0.149
BMI	0.829	0.028	0.015
Sitting height	0.770	−0.054	−0.081
Chest circumference	0.818	−0.054	−0.065
Upper arm circumference	0.823	0.022	−0.030
Shoulder width	0.671	−0.083	−0.139
Pelvis breadth	0.899	−0.014	−0.077
SBP	0.408	0.251	0.787
DBP	0.234	0.285	0.862
GLU	0.170	−0.020	0.016
TG	0.315	−0.057	−0.117
CHOL	0.057	0.950	−0.273
HDL-C	−0.010	0.338	−0.028
LDL-C	0.005	0.906	−0.237

**Table 6 nutrients-14-03095-t006:** The association of metabolic risk patterns with plasma and erythtocyte DHA status by multivariate logistic regression analysis.

Metabolic Risk Patterns	Plasma		Erythrocyte	
	OR (95% CI)	*p*	OR (95% CI)	*p*
Obesity risk pattern				
Model 1	0.870 (0.793, 0.953)	0.003	0.967 (0.931, 1.004)	0.079
Model 2	0.910 (0.825, 1.004)	0.060	0.968 (0.929, 1.008)	0.116
Model 3	0.873 (0.786, 0.969)	0.011	0.962 (0.923, 1.004)	0.075
Blood lipid risk pattern				
Model 1	1.276 (1.157, 1.406)	<0.001	1.047 (1.008, 1.088)	0.017
Model 2	1.288 (1.162, 1.428)	<0.001	1.046 (1.006, 1.088)	0.025
Model 3	1.271 (1.142, 1.415)	<0.001	1.043 (1.002, 1.086)	0.040
Blood pressure risk pattern				
Model 1	0.961 (0.878, 1.052)	0.391	0.978 (0.942, 1.016)	0.252
Model 2	0.946 (0.861, 1.040)	0.249	0.977 (0.940, 1.015)	0.238
Model 3	0.973 (0.880, 1.075)	0.585	0.983 (0.945, 1.023)	0.397

The results show OR (95% CI) of Q4 vs. Q1 by multivariate logistic regression analysis, with Q1 as the reference. Model 1: unadjusted; model 2: adjusted for age, sex, caregiver, caregiver’s education and occupation, and family economic level; model 3: adjusted for model 2 and intake of meat, poultry, eggs, and fish.

## Data Availability

The data presented in this study are available on request from the corresponding author. The data are not publicly available due to privacy.

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
