# Peer review of "Docosahexaenoic Acid as the Bidirectional Biomarker of Dietary and Metabolic Risk Patterns in Chinese Children: A Comparison with Plasma and Erythrocyte"

_nutrients, 2022, doi:10.3390/nu14153095_

Round 1

Reviewer 1 Report

This research evaluates the association between docosahexaenoic acid (DHA) in both plasma and erythrocyte with dietary and metabolic risk patterns in a sample of 435 Chinese children aged 5 to 7 years old.

Traditional diet assessment methods for people's intake rely on the accuracy of memories. There is the same problem in quantifying DHA. In addition, the information about the DHA content of some foods is not always available.

Having markers that allow us to identify dietary patterns and metabolic risk could be of great interest. However, the authors should clarify some methodological aspects of this study.

Specific comments

Introduction: 

-    Ln 45: “Several previous studies had confirmed that DHA are associated with many diseases, such as metabolic [4], immune [5], central nervous system disorders [6], and cancers [7].” Authors should clarify what they refer to with "DHA": DHA intake, DHA plasma or erythrocyte level? Correct the spelling: DHA are/is associated

-        Ln 59: “Whereas, few studies reported the associations of dietary patterns with DHA biomarker”. Authors should include references to these studies.

-      Ln 66: “metabolic risk indexes (e.g. blood glucose, blood lipid, blood pressure, or obesity)”. Authors should clarify what they want to say with: "metabolic risk indexes". Blood glucose, blood lipid and blood pressure are variables or indicators. Indices are complex indicators that summarize a set of variables or indicators.

-    Ln 74: Correct the spelling: may/might be an/a more excellent candidate biomarker to explore the assocaition/association.

Materials and Methods

-     Ln 94: “Diet information was collected using a validated food frequency questionnaire (FFQ)”. Please, add the reference of the questionnaire used.

-      Ln 108: “The detailed procedure and results of dietary patterns analysis had been reported in our previous study”. This information is crucial to understanding this study and should be summarized in this work to avoid looking for the other study to understand this one.

-      Ln 140: “Metabolic risk indices including weight, height, sitting height, chest circumference, upper arm circumference, shoulder width, pelvis breadth, body mass index (BMI), blood pressure (BP), blood glucose (GLU), triglyceride (TG), cholesterol (CHOL), high-density lipoprotein cholesterol (HDL-C), and low-density lipoprotein cholesterol (LDL-C) levels, were assessed in this study.” All of these are variables, they are not indices. The only indices used is BMI.  “The metabolic risk patterns were identified by principal component analysis referring to dietary pattern analysis. Factor scores were calculated and further categorized into four quartiles (Q1 to Q4)”. This information is crucial to understanding this study and should be summarized in this work.

Statistical analysis

-        Ln 170: replace B with beta (β)

Results

-    Ln 184-189: the majority of this information is included in Table 1. Avoid repetition.

-   Ln 196: Authors mentioned that this study aimed to measure docosahexaenoic acid (DHA) in both plasma and erythrocyte of the same children population and compared them with respect to their dietary patterns. Why are they showing in Table 2 the correlations of food items separately?

-      Ln 238: In the same way: The authors mentioned that the objective of this study was "to measure docosahexaenoic acid (DHA) in both plasma and erythrocyte of the same children population and compared them concerning their metabolic risk patterns". Then, why are they showing in Table 4 the correlations of the different variables included separately?

-      Ln 251: "Three metabolic risk patterns were identified in this study" The method used to identify these patterns is not clear. Please clarify. Please clarify Table 5.

Author Response

Reviewer 1

Comments and Suggestions for Authors

This research evaluates the association between docosahexaenoic acid (DHA) in both plasma and erythrocyte with dietary and metabolic risk patterns in a sample of 435 Chinese children aged 5 to 7 years old.

Traditional diet assessment methods for people's intake rely on the accuracy of memories. There is the same problem in quantifying DHA. In addition, the information about the DHA content of some foods is not always available.

Having markers that allow us to identify dietary patterns and metabolic risk could be of great interest. However, the authors should clarify some methodological aspects of this study.

Specific comments

Introduction: 

-    Ln 45: “Several previous studies had confirmed that DHA are associated with many diseases, such as metabolic [4], immune [5], central nervous system disorders [6], and cancers [7].” Authors should clarify what they refer to with "DHA": DHA intake, DHA plasma or erythrocyte level? Correct the spelling: DHA are/is associated

Respond: We are very sorry for our poor expression. We had corrected this sentence and the spelling. This sentence had been changed into “Several previous studies had confirmed that intake or circulating levels of DHA are associated with children’s health, such as metabolic[6], immune[7], and central nervous system disorders[8].”

-        Ln 59: “Whereas, few studies reported the associations of dietary patterns with DHA biomarker”. Authors should include references to these studies.

Respond: We are very sorry for our negligence. We had added relative references to these studies.

19.Tian, H.M.; Wu, Y.X.; Lin, Y.Q.; Chen, X.Y.; Yu, M.; Lu, T.; Xie, L. Dietary patterns affect maternal macronutrient intake levels and the fatty acid profile of breast milk in lactating Chinese mothers. Nutrition 2019, 58, 83-88, doi:10.1016/j.nut.2018.06.009.

  1. Benaim, C.; Freitas-Vilela, A.A.; Pinto, T.J.P.; Lepsch, J.; Farias, D.R.; Dos Santos Vaz, J.; El-Bacha, T.; Kac, G. Early pregnancy body mass index modifies the association of pre-pregnancy dietary patterns with serum polyunsaturated fatty acid concentrations throughout pregnancy in Brazilian women. Matern Child Nutr 2018, 14, doi:10.1111/mcn.12480.

-      Ln 66: “metabolic risk indexes (e.g. blood glucose, blood lipid, blood pressure, or obesity)”. Authors should clarify what they want to say with: "metabolic risk indexes". Blood glucose, blood lipid and blood pressure are variables or indicators. Indices are complex indicators that summarize a set of variables or indicators.

Respond: We are very sorry for our wrong expression. “metabolic risk indexes” was changed into“metabolic risk variables” in our manusript.

-    Ln 74: Correct the spelling: may/might be an/a more excellent candidate biomarker to explore the assocaition/association.

Respond: We are very sorry for our wrong expression. We had corrected this sentence.

Other studies reported that DHA in erythrocyte might be a more excellent candidate biomarker to explore the association with human diseases than other blood fractions.

Materials and Methods

-     Ln 94: “Diet information was collected using a validated food frequency questionnaire (FFQ)”. Please, add the reference of the questionnaire used.

Respond: We are very sorry for our negligence. We had added relative reference.

  1. Huang, Z.; Yin, X.-c.; Chen, M.; Li, M.-l.; Chen, B.; Hu, Y.-m. Relationships Between Dietary Patterns and Low-Level Lead Exposure Among Children from Hunan Province of China. Exposure and Health 2021, 10.1007/s12403-021-00432-6, doi:10.1007/s12403-021-00432-6.

-      Ln 108: “The detailed procedure and results of dietary patterns analysis had been reported in our previous study”. This information is crucial to understanding this study and should be summarized in this work to avoid looking for the other study to understand this one.

Respond: We are very sorry for our negligence. According to reviewer’s suggestion, we further summarized the detailed procedure and results of dietary patterns analysis. Factor loadings for dietary patterns of children in this study were added as supplement tables 1. The corrected results were showed in section of 2.2

-      Ln 140: “Metabolic risk indices including weight, height, sitting height, chest circumference, upper arm circumference, shoulder width, pelvis breadth, body mass index (BMI), blood pressure (BP), blood glucose (GLU), triglyceride (TG), cholesterol (CHOL), high-density lipoprotein cholesterol (HDL-C), and low-density lipoprotein cholesterol (LDL-C) levels, were assessed in this study.” All of these are variables, they are not indices. The only indices used is BMI.  “The metabolic risk patterns were identified by principal component analysis referring to dietary pattern analysis. Factor scores were calculated and further categorized into four quartiles (Q1 to Q4)”. This information is crucial to understanding this study and should be summarized in this work.

Respond: We are very sorry for our poor expression. According to reviewer’s suggestion, “metabolic risk indices” was corrected to “metabolic risk variables”. The procedure of metabolic risk patterns analysis was summarized in section 2.4.

Statistical analysis

-        Ln 170: replace B with beta (β)

Respond: We are very sorry for our improper expression. According to reviewer’s suggestion, B was replaced with β in multivariate linear regression analysis of our manusript.

Results

-    Ln 184-189: the majority of this information is included in Table 1. Avoid repetition.

Respond: We are grateful for this advice. According to reviewer’s suggestion, we delete the description for demographic characteristics and metabolic risk variables in section 3.1, which had been showed in Table 1.

-   Ln 196: Authors mentioned that this study aimed to measure docosahexaenoic acid (DHA) in both plasma and erythrocyte of the same children population and compared them with respect to their dietary patterns. Why are they showing in Table 2 the correlations of food items separately?

Respond: Firstly, we are grateful for this advice. Dietary pattern analysis by principal component analysis represented comprehensive diet from different food items to explore the associations of circulating levels of DHA with dietary patterns. The correlation of food items with plasma and erythrocyte DHA status can help us to further to speculate contribution of individual food in dietary pattern to circulating levels of DHA.

-      Ln 238: In the same way: The authors mentioned that the objective of this study was "to measure docosahexaenoic acid (DHA) in both plasma and erythrocyte of the same children population and compared them concerning their metabolic risk patterns". Then, why are they showing in Table 4 the correlations of the different variables included separately?

Respond: The reasons were consistent with the above.

-      Ln 251: "Three metabolic risk patterns were identified in this study" The method used to identify these patterns is not clear. Please clarify. Please clarify Table 5.

   Respond: We are very sorry for our improper expression. According to reviewer’s suggestion, the procedure of metabolic risk patterns analysis was summarized in section 2.4.

Reviewer 2 Report

In this manuscript entitled "Docosahexaenoic Acid as the Bidirectional Biomarker of Dietary and Metabolic Risk Patterns in Chinese Children:Comparison with Plasma and Erythrocyte", the authors evaluate docosahexaenoic acid (DHA) in both plasma and erythrocyte of the same children population and compared them with respect to their associations with dietary and metabolic risk patterns. Interesting data is presented. The authors have performed various analyses in this MS, but it is unclear how they performed them. I also think it is insufficient to show that DHA is a marker of both dietary and metabolic risk, as the authors claim in this MS. In this case, DHA is a good biomarker because the subjects were children from areas with low fish intake, but I think it would be difficult to use it in other areas. I think the authors need to be more limited in presenting their claims. I have comments, explained below. I hope that my comments are very useful for the improvement of this research.

Comments
(1)    Throughout: Why did authors focus only on DHA in this study? n-3PUFAs include ALA and EPA in addition to DHA. I think it is necessary to state the reason for focusing on DHA only.
(2)    Same children population: The authors refer to subjects as "the same group of children," but for what reason do they use the word "same"?
(3)    Abstract: I think authors should be able to understand the content of the MS to some extent just by reading the abstract. Therefore, authors should pay attention to the use of terminology. For example, a reader cannot tell what “diversified pattern” and “obesity risk pattern” stand for just by reading the abstract.
(4)    L32-33: Wording should be used with caution. This study is a cross-sectional study. Therefore, it is incorrect to say that the “Diversified dietary patterns improved plasma DHA level”. The correct phrase is: "Diversified dietary patterns had higher plasma DHA concentrations.
(5)    L44-46: Are the references in 4-7 references about children? Based on the previous sentence, it would be preferable to show the relationship between DHA and disease in children here.
(6)    L88: Are you fasting before blood draw? If not fasting, plasma DHA levels may be affected by diet.
(7)    L139: The authors have created an index called the Metabolic Risk Index. However, it is unclear what calculations were performed to obtain these indices. Please describe the calculation process in detail.
(8)    L152: Please also provide details on how the factor score was obtained.
Figures 3, 4: Figure 3 and 4 has larger text.
(9)    Table 3: What does B in Table 3 represent?
(10)    L251: Authors write "Three metabolic risk patterns were identified in this study", how was this confirmed? There are no details of this.
(11)    L285-286: Authors write " Many studies have assessed DHA status in different tissues, but comparisons of diverse tissues in the same population were rarely conducted", I think it is harsh to use the word "diverse" when authors have just measured plasma and red blood cells. Looking at other papers, it seems to be common practice to measure these two things (plasma and erythrocytes). Therefore, I do not think this reason gives this study an advantage.
(12)    L301: Why was there a strong correlation with eggs? Please discuss the reasons for this.
(13)    L355-357: What is used as an indicator to determine if the food quality is good or bad?

Author Response

Reviewer 2

Comments and Suggestions for Authors

In this manuscript entitled "Docosahexaenoic Acid as the Bidirectional Biomarker of Dietary and Metabolic Risk Patterns in Chinese Children:Comparison with Plasma and Erythrocyte", the authors evaluate docosahexaenoic acid (DHA) in both plasma and erythrocyte of the same children population and compared them with respect to their associations with dietary and metabolic risk patterns. Interesting data is presented. The authors have performed various analyses in this MS, but it is unclear how they performed them. I also think it is insufficient to show that DHA is a marker of both dietary and metabolic risk, as the authors claim in this MS. In this case, DHA is a good biomarker because the subjects were children from areas with low fish intake, but I think it would be difficult to use it in other areas. I think the authors need to be more limited in presenting their claims. I have comments, explained below. I hope that my comments are very useful for the improvement of this research.

Comments
(1)    Throughout: Why did authors focus only on DHA in this study? n-3PUFAs include ALA and EPA in addition to DHA. I think it is necessary to state the reason for focusing on DHA only.

Respond: We are grateful for this advice. According to reviewer’s suggestion, we added supplementary description of the reason for focusing on DHA only in the first paragraph.

(2)    Same children population: The authors refer to subjects as "the same group of children," but for what reason do they use the word "same"?

Respond: We are very sorry for our improper expression. We used the word "same" because we considered the samples of plasma and erythrocyte from the same children in last manusript. According to reviewer’s suggestion, we removed the word "same" in this manusript.

(3)    Abstract: I think authors should be able to understand the content of the MS to some extent just by reading the abstract. Therefore, authors should pay attention to the use of terminology. For example, a reader cannot tell what “diversified pattern” and “obesity risk pattern” stand for just by reading the abstract.

Respond: We are very sorry for our improper expression. According to reviewer’s suggestion, we added the description of dietary and metabolic risk patterns in section of abstract. “Diversified pattern” was revised into “Diversified dietary pattern characterized high intakes of diversified foods”, “obesity risk pattern” was revised into “obesity risk pattern with high weight, pelvis breadth, BMI, upper arm circumference, chest circumference”, and “blood lipid risk pattern” was revised into “blood lipid risk pattern with high CHOL and LDL-C level”.

(4)    L32-33: Wording should be used with caution. This study is a cross-sectional study. Therefore, it is incorrect to say that the “Diversified dietary patterns improved plasma DHA level”. The correct phrase is: "Diversified dietary patterns had higher plasma DHA concentrations.

Respond: We are very sorry for our improper expression. We revised these sentences in section of abstract and conclusion.

(5)    L44-46: Are the references in 4-7 references about children? Based on the previous sentence, it would be preferable to show the relationship between DHA and disease in children here.

Respond: According to reviewer’s suggestion, the references in 4-7 in last manusript were replaced by references about children in 6-8. 

  1. Jauregibeitia, I.; Portune, K.; Gaztambide, S.; Rica, I.; Tueros, I.; Velasco, O.; Grau, G.; Martin, A.; Castano, L.; Larocca, A.V., et al. Molecular Differences Based on Erythrocyte Fatty Acid Profile to Personalize Dietary Strategies between Adults and Children with Obesity. Metabolites 2021, 11, doi:10.3390/metabo11010043.
  2. Mikkelsen, A.; Galli, C.; Eiben, G.; Ahrens, W.; Iacoviello, L.; Molnar, D.; Pala, V.; Rise, P.; Rodriguez, G.; Russo, P., et al. Blood fatty acid composition in relation to allergy in children aged 2-9 years: results from the European IDEFICS study. Eur J Clin Nutr 2017, 71, 39-44, doi:10.1038/ejcn.2016.158.
  3. Baumgartner, J.; Smuts, C.M.; Malan, L.; Kvalsvig, J.; van Stuijvenberg, M.E.; Hurrell, R.F.; Zimmermann, M.B. Effects of iron and n-3 fatty acid supplementation, alone and in combination, on cognition in school children: a randomized, double-blind, placebo-controlled intervention in South Africa. Am J Clin Nutr 2012, 96, 1327-1338, doi:10.3945/ajcn.112.041004.

(6)    L88: Are you fasting before blood draw? If not fasting, plasma DHA levels may be affected by diet.

Respond: We are very sorry for our negligence. Venous whole blood samples after overnight fasting were collected in this study. We had revised the description in section 2.3.

(7)    L139: The authors have created an index called the Metabolic Risk Index. However, it is unclear what calculations were performed to obtain these indices. Please describe the calculation process in detail.

Respond: We are very sorry for our wrong expression. No metabolic risk index were created in this study. Only BMI index was caculated. “Metabolic Risk Index” in line 139 was revised into “Metabolic risk variables”. The assessment method of these metabolic risk variables had been incroduced in section 2.4.

(8)    L152: Please also provide details on how the factor score was obtained.
Figures 3, 4: Figure 3 and 4 has larger text.

Respond: We are very sorry for our negligence. We summarized the detail procedure of metabolic risk patterns analysis in section 2.4. The number of metabolic risk patterns were identified referring to dietary pattern analysis. Factor loadings represented the correlation between metabolic risk variables and metabolic risk patterns. Factor loadings >|0.3| were considered to contribute significantly to the metabolic risk patterns. Factor scores summed each metabolic risk variable weighted according to factor loadings, were calculated and further categorized into four quartiles (Q1 to Q4). Q1 represented a weak association with the metabolic risk pattern and Q4 represented a strong association with the metabolic risk pattern. The factor score was obtained had been introduced in section 2.2. Factor scores summed intakes of each variable group weighted according to factor loading, were calculated for each pattern and each individual.

(9)    Table 3: What does B in Table 3 represent?

Respond: We are very sorry for our negligence. Unstandardized coefficients (B) was replaced by standardized coefficients (β) to determine the strength of associations in multivariate linear regression analyses, which had been introduced in section 2.6. 

(10)    L251: Authors write "Three metabolic risk patterns were identified in this study", how was this confirmed? There are no details of this.

Respond: We are very sorry for our negligence. We summarized the detail procedure of metabolic risk patterns analysis in section 2.4.

(11)    L285-286: Authors write " Many studies have assessed DHA status in different tissues, but comparisons of diverse tissues in the same population were rarely conducted", I think it is harsh to use the word "diverse" when authors have just measured plasma and red blood cells. Looking at other papers, it seems to be common practice to measure these two things (plasma and erythrocytes). Therefore, I do not think this reason gives this study an advantage.

Respond: We are very sorry for our improper expression. We removed the improper description in first paragraph of discussion. Moreover, we revised the description of strengths and limitations in last paragraph of discussion.

The strength of this study is that “This is the first study reporting the comparative assessment of plasma and eryth-rocytes DHA as bidirectional biomarkers of dietary and metabolic risk patterns in Chinese children” . In addition, we further pointed out the limitation. “Lastly, majority of studies reported circulating levels of DHA were good biomarkers of marine foods intake[52,53]. In this case, the subjects were children from areas with low marine foods intake, the association of plasma DHA with dietary pattern might be difficult to use it in children from other areas.”

(12)    L301: Why was there a strong correlation with eggs? Please discuss the reasons for this.

Respond: According to reviewer’s suggestion, we added discussion about the reason why there a strong correlation with eggs.

(13)    L355-357: What is used as an indicator to determine if the food quality is good or bad?

Respond: We are very sorry for our improper expression. The conclusion had been rewritten. The relative sentence was changed into “Plasma DHA may be a more sensitive bidirectional biomarker to evaluate recently comprehensive diet intake and metabolic risk of children”.

Round 2

Reviewer 2 Report

I am satisfied with the revisions that have been made by the authors.